# The Spatiotemporal Patterns of Climate Asymmetric Warming and Vegetation Activities in an Arid and Semiarid Region

**Tong Heng** [1], **Gary Feng** [2,*], **Ying Ouyang** [3] **and Xinlin He** [1]

1    College of Water and Architectural Engineering, Shihezi University, Shihezi 832021, China;
     htshz121@163.com (T.H.); hexinlin2002@163.com (X.H.)
2    USDA-ARS, Genetic and Sustainable Agricultural Research Unit, 810 Hwy 12 East,
     Mississippi State University, Starkville, MS 39762, USA
3    USDA Forest Service, Center for Bottomland Hardwoods Research, 775 Stone Blvd., Thompson Hall,
     Room 309, Mississippi State University, Starkville, MS 39762, USA; ying.ouyang@usda.gov
*    Correspondence: gary.feng@usda.gov; Tel.: +1-1662-320-7449

**Abstract:** Asymmetric warming was bound to have a major impact on terrestrial ecosystems in arid regions during global warming. Further study was necessary to reveal the spatiotemporal patterns of asymmetric warming in Xinjiang; this study analyzed the climate and normalized difference vegetation index (NDVI) data (2000–2020). The change trends of the day and nighttime warming (DNW), seasonal warming, and the diurnal temperature range in northern Xinjiang (S1) and southern Xinjiang (S2) were determined. The findings indicated that the DNW rate showed a significant ($p < 0.05$) upward trend, especially in winter. The nighttime warming rate (0.65 °C (decade)$^{-1}$) was faster than the daytime warming rate (0.4 °C (decade)$^{-1}$), and the diurnal temperature range between daytime and nighttime exhibited a decreasing trend. The diurnal temperature range was the highest in spring and the lowest in winter. Extreme values of the diurnal temperature range appeared in autumn (48.6 °C) and winter (12.3 °C) and both in S1. The $T_{min}$ in S1 had an abruption trend in 2006–2017, the $T_{max}$ in S2 had an abruption trend in 2005–2011, and the probability of spatial abruption in S1 was higher than that in S2. The partial correlation between the NDVI and $T_{min}$ was significantly higher than that between the NDVI and $T_{max}$ in the area where the significance test passed; therefore, asymmetric nighttime warming had a greater impact on the NDVI than the asymmetric daytime warming.

**Keywords:** asymmetric warming; normalized difference vegetation index; second-order partial correlation analysis; day and nighttime warming; diurnal temperature range

## 1. Introduction

Since the 1950s, terrestrial ecosystems have experienced a continuous warming process [1–3]. With the increase in global average temperature, there is growing evidence suggesting that asymmetric patterns of day and nighttime warming (DNW) and seasonal warming are common in the warming process [4–7]. This phenomenon is called asymmetric warming [8]. The global average temperature has risen by 0.89 °C over the last 100 years [9], and the rate of temperature increase is 0.13 °C (decade)$^{-1}$ [10,11]. Average global temperatures are projected to increase by 0.91–2 °C throughout the mid-21st century [12,13]. Located in the eastern part of Eurasia, China is one of the most complex regions experiencing global climate change and is strongly influenced by the eastern continental monsoon climate and the northwest inland arid climate [14,15]. In the past 40 years, the average surface temperature in China has risen by 1.1 °C, which is higher than the global warming rate [16–20].

The persistent enhancement of asymmetric warming will have significant impacts on terrestrial vegetation and even on the whole structure, functions, and services of ecosystems.

The IPCC's (Intergovernmental Panel on Climate Change) fourth assessment reported that the global nighttime warming rate from 1957 to 2007 was 1.4 times the daytime warming rate [21,22]. Davy et al. [23] showed that the warming rate in summer was faster than that in spring and autumn in the high latitudes of the Northern Hemisphere. As an important part of terrestrial ecosystems, the growth and development of vegetation are bound to be affected by asymmetric warming. These generally relate differences in the temperature trends to regionalized cloud cover, precipitation, or soil moisture. Peng et al. [24] analyzed the interannual covariations of the satellite-derived normalized difference vegetation index (NDVI) with asymmetric warming over the Northern Hemisphere; their study showed that daily maximum temperatures were positively correlated with the NDVI in humid areas and had a significant negative correlation with the NDVI in arid areas. Xu et al. [25] investigated vegetated growth dynamics (annual productivity, seasonality, and the minimum amount of vegetated cover) in China and their relations with climatic factors during 1982–2011, and they believed that vegetation productivity was positively correlated with the nighttime warming rate. Nemani et al. [26] analyzed the climatic data and satellite observations of vegetation activity during 1982–1999 and found that vegetation growth representing the response cycle of asymmetric warming in high-latitude and high-altitude areas was shorter than that in low-latitude and low-altitude areas. Cong et al. [27] showed that asymmetric warming could accelerate respiration in plants and increase the decomposition rate of organic matter. However, little is known about the warming trends in the relationships between NDVI and asymmetric warming and precipitation, and understanding this is crucial for predicting how climate change would affect vegetation activity in the future.

Located in the arid and semiarid region of China, with a temperate continental climate, Xinjiang is a typical "mountain-oasis-desert" ecosystem [28]. As the area is affected by the sea-land distribution and the uplift of the Qinghai-Tibet Plateau, the differences in soil water-thermal status and geographical vegetation differentiation between S1 and S2 have been significant; these differences have led to different sensitivities of the vegetation types to climate factors in Xinjiang [29,30]. At present, research on the responses of vegetation activities to asymmetric warming mainly focuses on the differences in interannual average temperature and the geographic spatial and seasonal changes in vegetation types [31–33]. The changing trends of asymmetric warming, abruption characteristics, and the influence of precipitation on the NDVI over different time series have received little attention. Given this, in the context of global asymmetric warming, this paper was based on a global NDVI dataset, monthly minimum ($T_{min}$) and maximum temperature ($T_{max}$) data, and precipitation data in Xinjiang from 2000 to 2020. Least squares linear regression, Yamamoto and Mann–Kendall nonparametric randomization tests, and second-order partial correlation analysis were used to study the spatiotemporal patterns and abruption signatures of daytime and nighttime warming, seasonal warming, the diurnal temperature range, and the response of vegetation activities to asymmetric warming. The goal of this study was to provide a reference for the impacts of asymmetric warming on the vegetation ecosystems in Xinjiang.

## 2. Materials and Methods

### 2.1. Site Description

The geomorphological features of Xinjiang (73°40′–96°23′ E and 34°25′–49°10′ N) consist of 3 mountains and 2 basins, with a west-east width of approximately 1950 km, south-north length of approximately 1550 km, and a total area of $1.66 \times 106$ km$^2$. The mountain and basins, from north to south, are the Altai Mountains, the Junggar Basin, the Tianshan Mountains, the Tarim Basin, and the Karakoram Mountains. The Tianshan Mountains lie across the middle of the region [34]. On the basis of the meteorological definition and geomorphological characteristics of Xinjiang, the study area is further categorized into two regions: northern Xinjiang (S1) and southern Xinjiang (S2), with the Tianshan Mountain as the boundary (Figure 1). The mean annual temperatures in S1 and S2 are 13 and

19 °C, and the annual precipitation values are 180 and 95 mm, respectively; these precipitation values are only 15–28% of the average precipitation level of China (630 mm, 2000–2020).

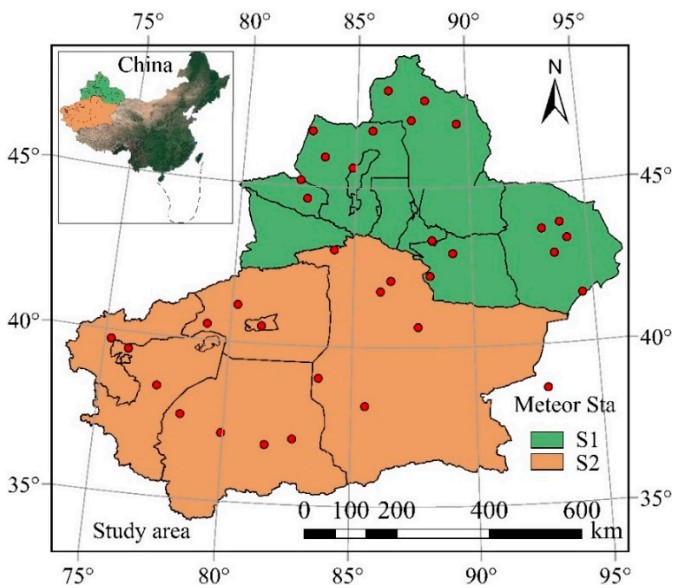

**Figure 1.** Distribution of meteorological stations in the studied region.

Meteorological data were obtained from the dataset of monthly surface climate data of the China Meteorological Science Data service network (http://cdc.cma.gov.cn). The weather data are the maximum daytime temperature ($T_{max}$), the minimum nighttime temperature ($T_{min}$), and precipitation. To ensure homogeneity and the longest continuous observation, data from 34 ground-based meteorological stations of the China Meteorological Administration (CMA) were used in the study. The Kriging interpolation method was chosen in ArcGIS 10.4 with the spatial analyst extension and raster images. According to the existing seasonal division method [35], the monthly dataset was further averaged to spring (March, April, and May), summer (June, July, and August), autumn (September, October, and November), winter (December, January, and February), the growing season (April to October), and the non-growing season (November to March of the subsequent year).

The NDVI dataset from 2000 to 2020 was acquired from the MOD13A3 product of Goddard Space Flight Center, LAADS, NASA (https://ladsweb.modaps.eosdis.nasa.gov/), and the spatiotemporal resolution is 1 km, 30 days, respectively. This dataset was synthesized by the maximum value synthesis method (MVS) and processed by cloud, aerosol, water vapor, etc.; it is the longest time series currently acquired by the TERRA satellite.

*2.2. Research Methods*

The least-square linear regression method was applied to analyze the trend of DNW at the regional and pixel scales [36,37], and the significance of regression coefficients was determined by the *t*-test (*p*-values). * indicated significant (*p* = 0.05). ** indicated extremely significant (*p* = 0.01). The diurnal warming rate was calculated element-by-element, and the spatiotemporal patterns of the DNW rates in different seasons were analyzed.

$$C_j = \frac{19 \times \sum_{i=1}^{19} i \cdot \varphi - \sum_{i=1}^{19} i \cdot \sum_{i=1}^{19} \varphi}{19 \sum_{i=1}^{19} i^2 - \left(\sum_{i=1}^{19} i\right)^2} \tag{1}$$

where *i* is the time series, ranging from 2000 to 2020; $\varphi$ is the specified factor, including NDVI, $T_{min}$, and $T_{max}$; $C_j$ is the change tendency rate of the $\varphi$ pixel in 20 years, $C_j > 0$ indicates that the $\varphi$ factor of

pixel $j$ has an increasing trend in the past 20 years, $C_j < 0$ indicates a decreasing trend, and $C_j = 0$ indicates no change.

Mann–Kendall and Yamamoto methods were employed to test the abruption of temperature in each time series [38,39]. Mann–Kendall method is a non-parametric statistical test. It is suitable for both type variable and ordinal variable, with no specific requirements on the distribution of samples, and Pearson correlation results were not dominated by outliers. The major mathematical methods were as follows:

For the time series, a rank s series is constructed:

$$S_k = \sum_{i=1}^{k} r_i \quad k = 2,3 \ldots n \tag{2}$$

$$r_i = \begin{cases} 1 \; x_i > x_j \\ 0 \; x_i < x_j \end{cases} \quad j = 1,2 \ldots i \tag{3}$$

where rank series $S_k$ is the cumulative number of the $i$ time value greater than $j$ time. The statistics was defined under the assumption that the time series were independent:

$$UF_k = \frac{\left| S_k - E(S_k) \right|}{\sqrt{var(S_k)}} \quad k = 1,2 \ldots n \tag{4}$$

$$UB_k = -UF_k \quad k = n, n-1 \ldots 1 \tag{5}$$

where $UF_1 = 0$, $E(S_k)$ and $var(S_k)$ are the mean and variance of $S_k$. When $x_1, x_2 \ldots x_n$ were independent of each other and had the same continuous distribution, they were calculated as follows:

$$\begin{cases} E(S_k) = \frac{k(x-1)}{4} \\ var(S_k) = \frac{k(x-1)(2k+5)}{72} \end{cases} \quad k = 2,3 \ldots n \tag{6}$$

Set $UF_k$ and $UB_k$ as standard normal distribution, given a significance value $\alpha = 0.05$, critical value $U_{0.05} = \pm 1.96$, and two statistical sequence curves of $UF_k$ and $UB_k$ and the two critical lines ($\pm 1.96$) were drawn in one graph.

Yamamoto method explores the abruption characteristics from two parts: climate information and climate noise. In this study, DPS 7.05 (http://www.chinadps.net/) was used to process the abruption data. The specific method was to define a signal-to-noise ratio:

$$\frac{S}{N} = \frac{\left| \overline{x_1} - \overline{x_2} \right|}{S_1 + S_2} \tag{7}$$

where $S$ is the population sample variance, and $N$ is the length of the population sample sequence; $\overline{x_1}$ and $\overline{x_2}$ are average values of 2 sub-sample sets; $S_1$ and $S_2$ are variance of sub-sample. Molecular was defined when $S/N > 1.0$.

To eliminate the interference of other variables except for temperature and precipitation, the second-order partial correlation analysis was used to investigate the influence of DNW on vegetation NDVI [39–41]. First, the second-order partial correlation between $T_{max}$ and NDVI was analyzed through limiting the influence of $T_{min}$ and precipitation; second, the second-order partial correlation between $T_{min}$ and NDVI was analyzed through limiting the influence of $T_{max}$ and precipitation. The second-order partial correlation coefficient was calculated based on the first-order partial correlation coefficient, and the first-order correlation coefficient needed to be calculated for

Pearson's correlation coefficient (zero-order). The partial correlation coefficients were calculated as follows:

$$r_{\alpha\delta} = \frac{\sum_{i=1}^{n}(\alpha_i - \overline{\alpha})(\varphi_i - \overline{\delta})}{\sqrt{\sum_{i=1}^{n}(\alpha_i - \overline{\alpha})^2 \sum_{i=1}^{n}(\delta_i - \overline{\delta})^2}} \tag{8}$$

$$r_{\alpha\delta.1} = \frac{r_{\alpha\delta} - r_{\alpha.1}r_{\delta.1}}{\sqrt{1 - r_{\alpha.1}^2} \cdot \sqrt{1 - r_{\delta.1}^2}} \tag{9}$$

$$r_{\alpha\delta.12} = \frac{r_{\alpha\delta.1} - r_{\alpha2.1}r_{\delta2.1}}{\sqrt{1 - r_{a2.1}^2} \cdot \sqrt{1 - r_{\delta2.1}^2}} \tag{10}$$

where $\alpha$ and $\varphi$ are principal variables by partial correlation coefficients; 1 and 2 are control variables; $r_{\alpha\delta}$ is correlation coefficient; $r_{\alpha\delta.1}$ is the first-order partial correlation coefficient; $r_{\alpha\delta.12}$ is the second-order partial correlation coefficient. *t*-test was used to test the significance of the calculated second-order partial correlation coefficient. The formula is as follows:

$$t = \frac{r\sqrt{n - q - 1}}{\sqrt{1 - r^2}} \tag{11}$$

where *r* is correlation coefficient; *n* and *q* are the number of samples and degrees of freedom.

## 3. Results

### 3.1. Interpreting Trends in Asymmetric Warming over Time

The interannual variation trend of DNW was calculated by a linear regression equation (Figure 2). Over the past 20 years, $T_{max}$ and $T_{min}$ in Xinjiang showed different degrees of warming trends. DNW had a local response period, i.e., cooling down every 2–3 years. The warming rates of $T_{max}$ in S1 and S2 were 0.5 and 0.3 °C (decade)$^{-1}$, respectively, but the overall $T_{max}$ in S1 was 3–5 °C lower than that in S2 ($p < 0.01$). The warming rates of $T_{min}$ in S1 and S2 were 0.8 and 0.5 °C (decade)$^{-1}$, respectively. From 2000–2020, the interannual $T_{min}$ in S1 increased from −7.1 to −4.3 °C, and the $T_{min}$ in S2 increased from −0.8 to 1.4 °C. Therefore, the interannual $T_{max}$ and $T_{min}$ in both S1 and S2 presented asymmetrical daytime and nighttime temperature warming, i.e., the nighttime warming rates (0.65 °C (decade)$^{-1}$) were faster than the daytime warming rates (0.40 °C (decade)$^{-1}$).

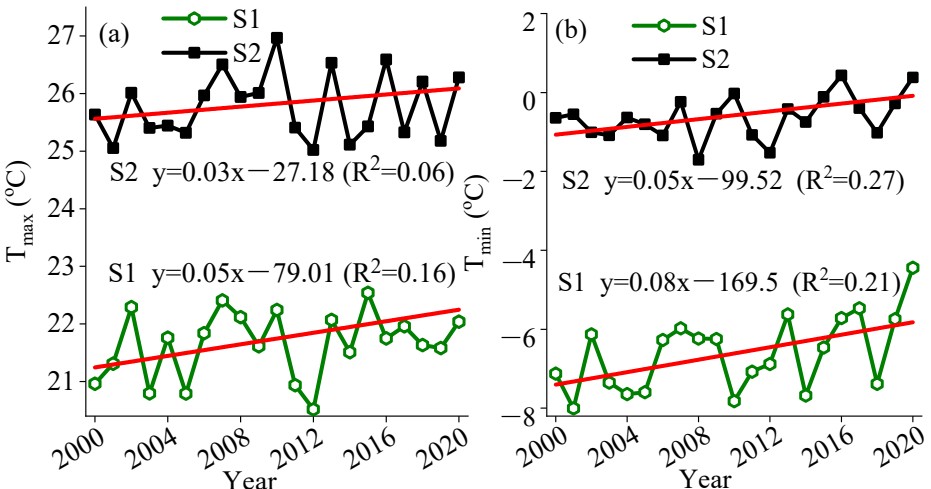

**Figure 2.** $T_{max}$ (**a**) and $T_{min}$ (**b**) trends in the S1 and S2 regions with arid and semiarid climate, 2000–2020.

There was considerable variation in the trends of seasonal $T_{max}$ and $T_{min}$ warming in S1 and S2 (Figure 3). The $T_{max}$ in S1 showed a cooling downward trend in summer and significant warming in other seasons. The $T_{max}$ in S2 showed a warming trend only in winter and a cooling down trend in other seasons. The $T_{min}$ in S1 showed a cooling down trend in autumn and significant increases in other seasons. The $T_{min}$ in S2 showed cooling downward trends in summer and autumn and significant warming trends in spring and winter. In addition, S1 and S2 had opposite temperature trends in spring $T_{max}$, summer $T_{min}$, and autumn $T_{max}$; in these three seasons, the $T_{max}$ and $T_{min}$ in S1 both showed a warming trend, while the temperatures in S2 showed a cooling trend. This phenomenon is mainly due to the latitude differences between S1 and S2, which further supports the significant warming trend in higher latitudes in the northern hemisphere.

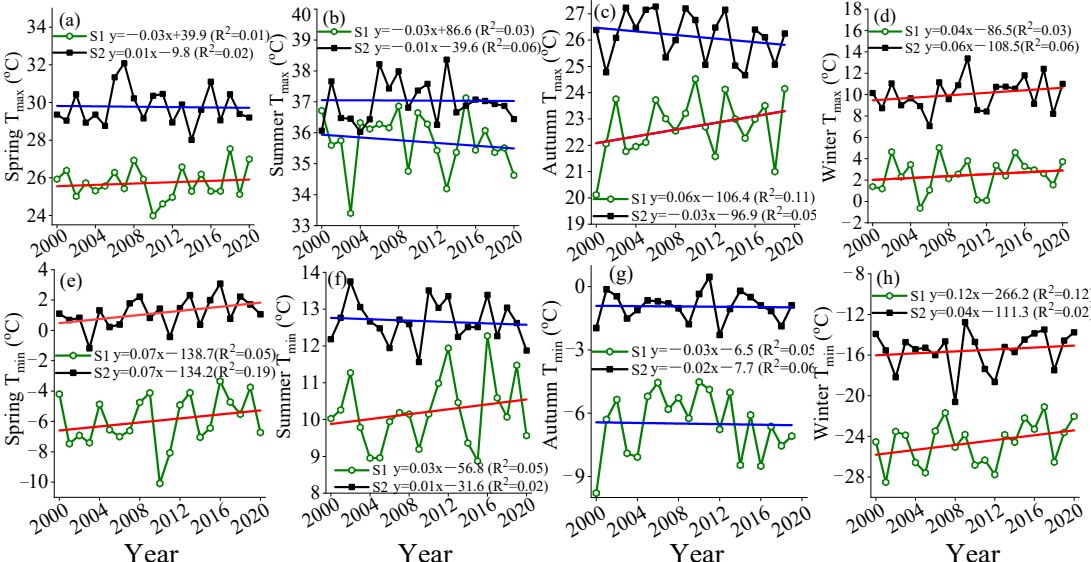

**Figure 3.** Seasonal $T_{max}$ and $T_{min}$ temperature trends in the studied regions S1 and S2, (**a**): spring $T_{max}$, (**b**): summer $T_{max}$, (**c**): autumn $T_{max}$, (**d**): winter $T_{max}$, (**e**): spring $T_{min}$, (**f**): summer $T_{min}$, (**g**): autumn $T_{min}$, (**h**): winter $T_{min}$. S1 and S2 represent two regions of the study area.

The DNW rates in S1 and S2 were asymmetrical in the four seasons. The warming rates of $T_{min}$ in winter and spring in S1 were 3 and 3.5 times those of $T_{max}$, respectively; S1 had higher nighttime warming trends than daytime warming trends in spring, summer, and winter, while the opposite trend was seen for autumn. The warming rates of $T_{min}$ in spring and autumn in S2 were higher than the $T_{max}$ warming trend; inversely, for the daytime warming trend, the warming rate of $T_{max}$ was higher than the $T_{min}$ warming trend in winter, and the warming rate of $T_{min}$ in winter in S2 was 0.6 times that of $T_{max}$. In addition, the warming rates of $T_{min}$ and $T_{max}$ were $-0.1$ °C (decade)$^{-1}$ in summer, which failed to pass the significance test at the 0.05 level; thus, the cooling trend was not significant. Comparing the DNW rates of the two study areas, the warming rates of $T_{max}$ in autumn in S1 and $T_{max}$ in winter in S2 were the same (0.6 °C (decade)$^{-1}$) and were significantly higher than those in other seasons ($p < 0.01$). The $T_{min}$ warming rate in S1 was fastest in winter (1.2 °C (decade)$^{-1}$), and the $T_{min}$ warming rate in S2 was fastest in spring (0.7 °C (decade)$^{-1}$).

There were regional differences in the diurnal temperature ranges in S1 and S2 in the four seasons (Figure 4). From 2000 to 2020, the diurnal temperature ranges in spring, summer, autumn, and winter in Xinjiang were 30.2, 25.3, 28.6, and 24.4 °C, respectively. The interannual diurnal temperature range showed a trend of spring > autumn > summer > winter. The diurnal temperature range in S1 was significantly higher than that in S2 in the spring and summer seasons in the second part of the time series (2013–2020); the trends in other seasons were not significant. The maximum and minimum diurnal temperature ranges appeared in autumn (48.6 °C) and winter (12.3 °C) in Xinjiang, and both

appeared in S1 areas. The maximum diurnal temperature range in S2 appeared in the winter of 2008 (47.8 °C).

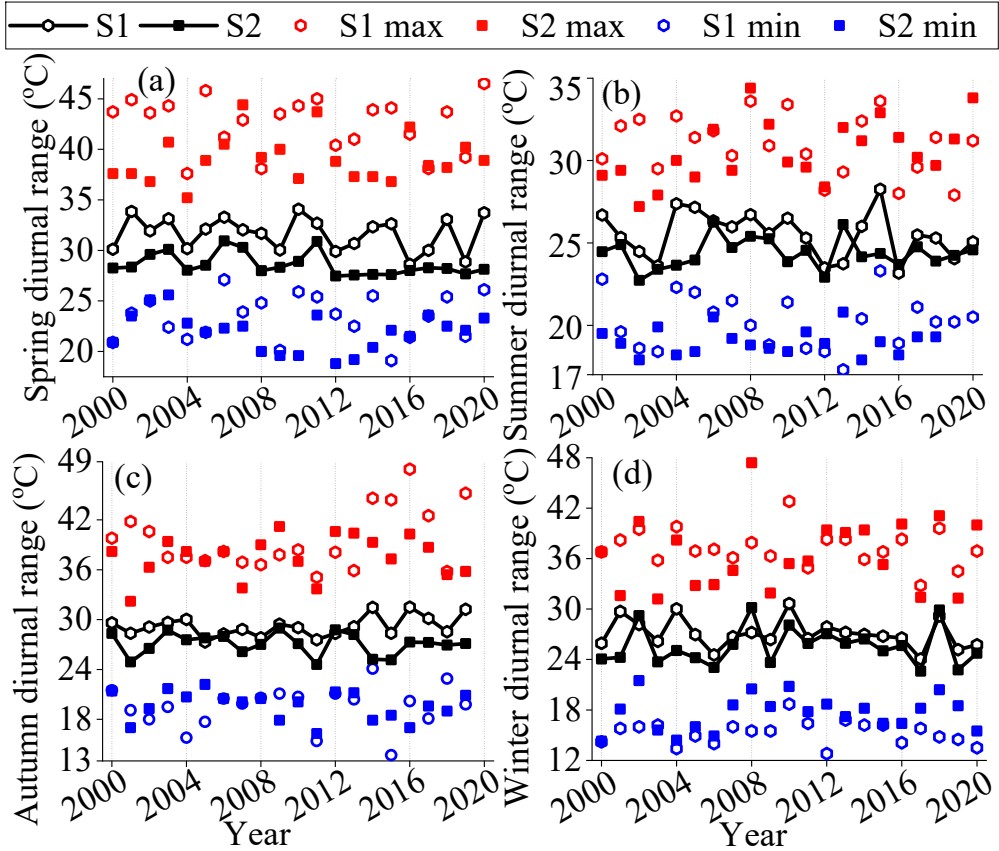

**Figure 4.** Seasonal variation trend of diurnal temperature range, (**a**): spring, (**b**): summer, (**c**): autumn, (**d**): winter. The red and blue scatter points represent the maximum and minimum of the daily range; S1 and S2 represent two regions of the study area.

## 3.2. Spatial Distribution Characteristics of Asymmetric Warming

The warming rates of $T_{max}$ and $T_{min}$ in Xinjiang from 2000 to 2020 were spatially inconsistent (Figure 5a,b). Specifically, the $T_{max}$ warming rate was between −0.04 and 2.19 °C $(decade)^{-1}$, and the $T_{min}$ warming rate was between −3.4 and 4.38 °C $(decade)^{-1}$. The DNW rates in S1 and S2 both showed asymmetric warming trends, and the DNW rate in S1 was significantly higher than that in S2. The $T_{max}$ and $T_{min}$ in S2 showed cooling trends in the southwest and central regions, and the proportions of cooling trends were 5.3% and 2.9%, respectively.

The warming rate of the diurnal temperature range in the growing season in Xinjiang showed asymmetric warming trends of −2.07 and 2.15 °C $(decade)^{-1}$ appeared in the western and eastern regions (Figure 5c), i.e., the trend was high in the eastern parts of the province, while in the western parts, the trend was low. The warming rate of the diurnal temperature range in the nongrowing season showed south- and north-low, middle-high trend (Figure 5d). In general, the diurnal temperature range in S2 showed a downtrend in the growing season, while that in S1 only showed a downtrend during the nongrowing season of vegetation.

## 3.3. Abruption Characteristics of DNW Warming

The abruption of DNW in Xinjiang was analyzed by the Yamamoto test (Figure 6). The changing point of $T_{max}$ warming in S1 and S2 occurred in 2006. The warming of $T_{min}$ in S1 had a strong abruption

in 2006 and a weak abruption in 2010, but the warming of $T_{min}$ in S2 did not reach the detection standard of abruption. Therefore, the probability of S1 abruption was overall higher than that of S2.

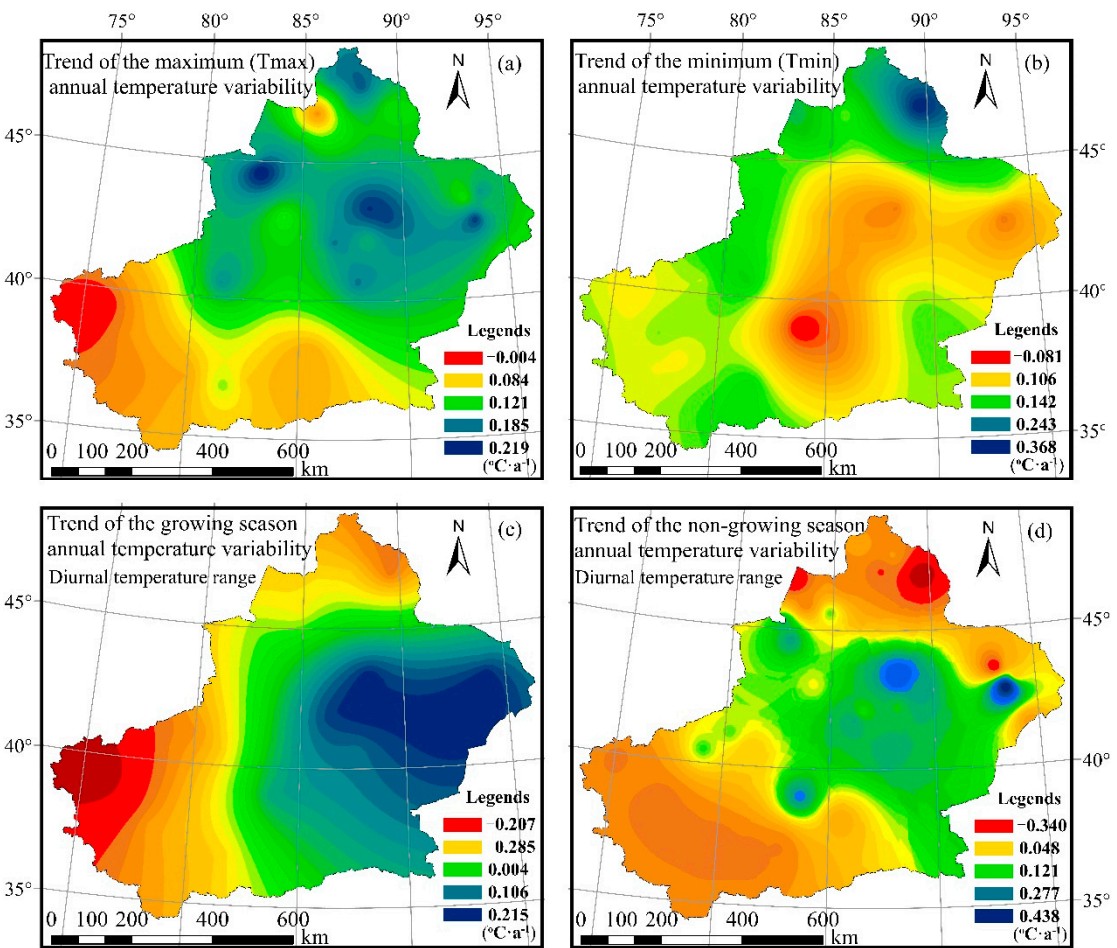

**Figure 5.** The spatial distribution of averaged $T_{max}$ and $T_{min}$ over 20 years in the arid and semiarid region, (**a**): the trend of $T_{max}$, (**b**): the trend of $T_{min}$, (**c**): diurnal range in the growing season of vegetations, (**d**): diurnal range in the non-growing season of vegetations. S1 and S2 represent two regions of the study area.

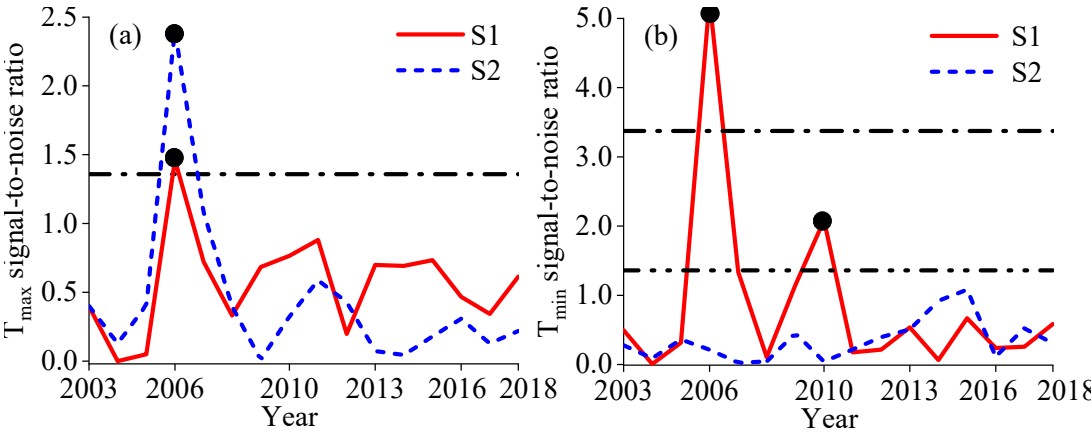

**Figure 6.** Abrupt change detection for (**a**): $T_{max}$, (**b**): $T_{min}$, Yamamoto test. S1 and S2 represent two regions of the study area.

The Mann–Kendall abruption test curve of DNW in Xinjiang was analyzed (Figure 7). Taking the critical value at the (±) 95% confidence level ($Z\alpha/2 = 1.96$) and a 5% degree of precision, it could be found from the UF and UB curves of the reverse time series, as well as their intersection point, that the warming trends of $T_{min}$ in S1 and S2 showed upward trends since 2003. Among the abruption results, S1 only presented a point abruption in 2017 ($p < 0.01$); the first intersection point of the UF and UB curves appeared in 2006, and this intersection point was within the confidence interval. Therefore, it was determined that the first abruption year of $T_{min}$ warming in S1 was 2006; the abruption then reached an extremely significant level in 2017. The warming trend of $T_{max}$ in S1 was not significant ($p$ for the trend = 0.103), while the warming trend of $T_{max}$ in S2 showed an upward trend in 2005–2011. The S2 region presented a point abruption in 2011 ($p < 0.01$); the first intersection point of the UF and UB curves also appeared in 2006, and this intersection point was within the confidence interval. Similarly, the abruption of $T_{max}$ warming in S2 started in 2005 and reached an extremely significant level in 2011.

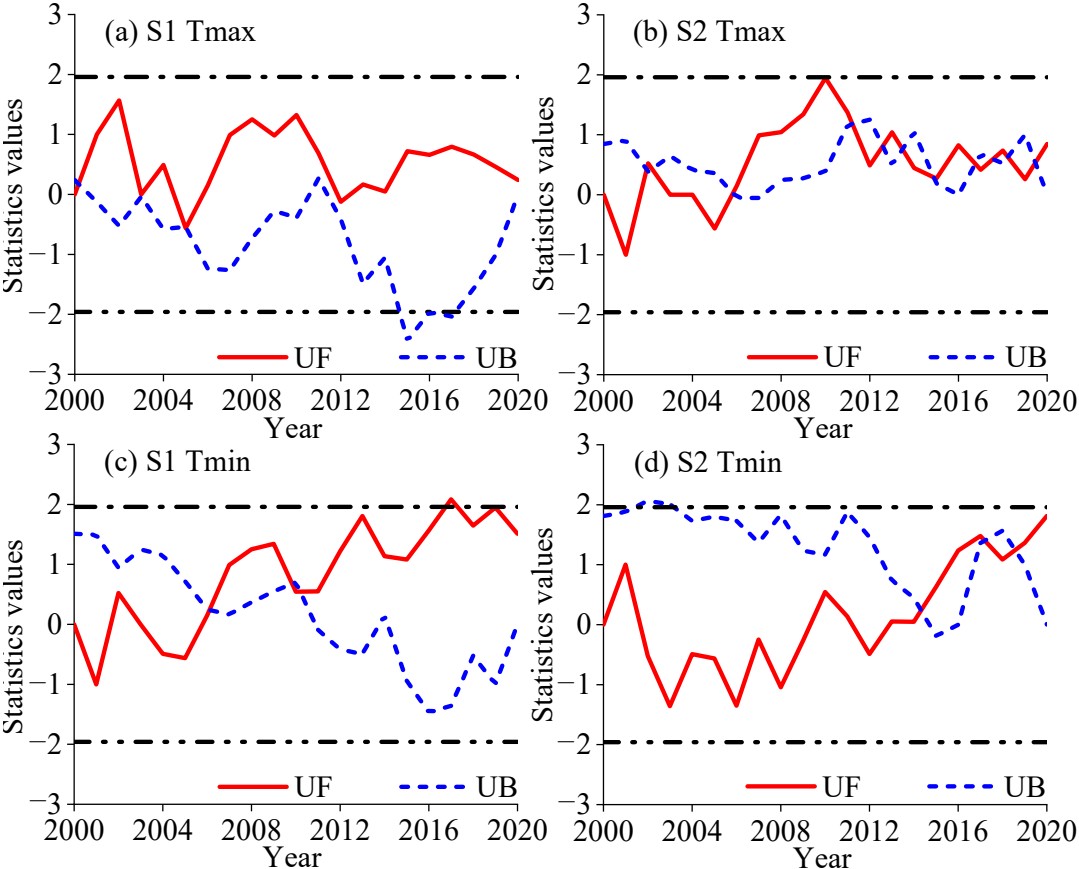

**Figure 7.** Mann–Kendall test for monotonic trend of (**a**) $T_{max}$ over S1 region, (**b**) $T_{max}$ over S1 region, (**c**) $T_{min}$ over S1 region, and (**d**) $T_{min}$ over S1 region. S1 and S2 represent two regions of the study area, UF and UB are statistics about temperature series, which obeys standard normal distribution. If UF and UB intersect, they represent the start time of an abrupt change.

*3.4. Partial Correlation Analysis between Vegetation Activity and Asymmetric DNW in Xinjiang*

Precipitation was included as a control variable, and partial correlation analysis of NDVI and DNW was carried out at 34 meteorological stations in Xinjiang (Figure 8). The area of a positive correlation between the NDVI and $T_{max}$ accounted for approximately 32.8% of the total pixels, of which 7.7% showed a significant positive correlation ($p < 0.05$). Approximately 68.1% of the studied area and $T_{max}$ were negatively correlated, and 4.7% of those pixels were significantly negatively correlated

($p < 0.05$). The negative correlation areas were mainly distributed in the central part of the S1 area (Junggar Basin).

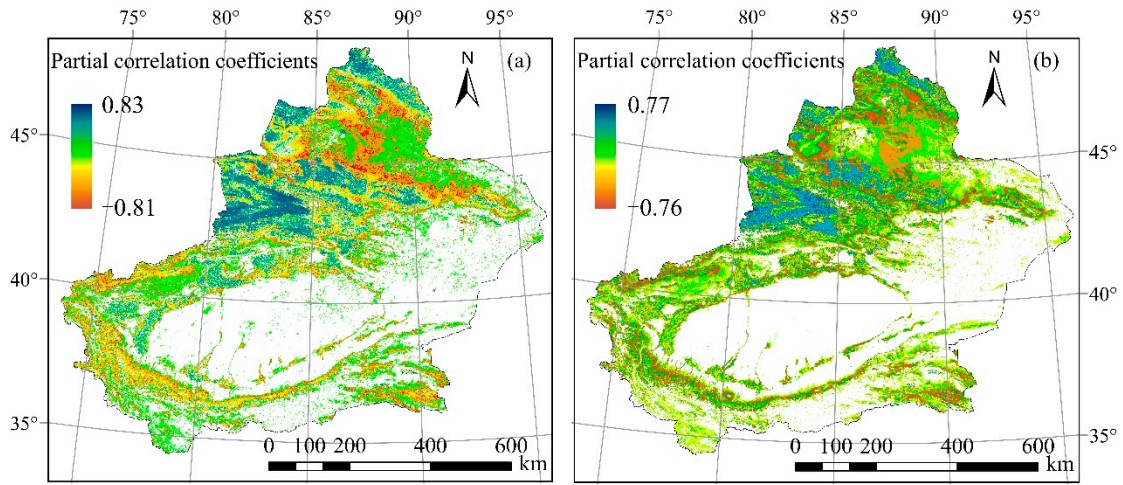

**Figure 8.** The partial correlation coefficient of vegetation NDVI (normalized difference vegetation index) with $T_{max}$ (**a**) and $T_{min}$ (**b**) in the arid and semiarid regions.

The distribution pattern of the correlation between NDVI and $T_{min}$ was opposite to that of NDVI and $T_{max}$; approximately 60.7% of the studied area and $T_{min}$ were positively correlated, of which 10.2% showed a significant positive correlation ($p < 0.05$). Approximately 39.1% of the studied area and $T_{min}$ were negatively correlated, and 6.1% of those pixels were significantly negatively correlated ($p < 0.05$). The negative correlation areas were mainly distributed in the north-central part of the S1 area (Gurbantünggüt Desert plain). Thus, the effect of nighttime warming on vegetation in Xinjiang was more extensive than the effect of daytime warming. From the spatial distribution of DNW, the distribution trend of the partial correlation coefficient between NDVI and DNW was basically consistent in the S2 area.

## 4. Discussion

### 4.1. Causes of Asymmetric Warming

The warming rates of $T_{max}$ and $T_{min}$ in Xinjiang from 2000 to 2020 showed significant upward trends. The warming rate of $T_{min}$ was 1.6 times (S1) to 1.67 times (S2) that of $T_{max}$. Li et al. [10] found that the summer $T_{max}$ and $T_{min}$ warming trends in Xinjiang from 1961 to 2005 were significant and that the interannual warming rate was 0.28 °C. The results of this study showed that the warming rates of $T_{max}$ and $T_{min}$ in the S2 area were 0.3 and 0.5 °C (decade)$^{-1}$, respectively, which were higher than those reported by Li et al. [10]. The warming rates of $T_{max}$ and $T_{min}$ further corroborated the research results. Tan et al. [5] showed that the global land surface temperature in the past 50 years increased significantly faster at nighttime than in the daytime and that the nighttime warming rate was 1.4 times that of the daytime. The DNW trends in S1 and S2 in this study had global coherence, and the warming rates in these areas were higher than the world average. However, the DNW was asymmetric; i.e., the nighttime warming rate (0.65 °C (decade)$^{-1}$) was faster than the daytime warming rate (0.4 °C (decade)$^{-1}$). The S1 study area has a temperate continental arid and semiarid climate with a large basin area, while S2 has a warm temperate continental arid climate and the world's second-largest mobile desert. The atmospheric inverse radiation in S2 is higher than that in S1, and the average annual precipitation in S1 is twice that in S2. These climate differences affect the atmospheric circulation of the two regions; thus, the difference in climate types was the main reason that the nighttime warming rate

was higher than the daytime warming rate [42,43]. Another possible explanation for this is the impact of human activities on DNW [44,45].

### 4.2. Abruption Trend Analysis of Asymmetric Warming

In Xinjiang, $T_{max}$ and $T_{min}$ experienced sudden warming in 2011 and 2017, respectively, and the warming trend of $T_{min}$ after 2005 was significant. There were also asymmetric changes in the abruption times of $T_{max}$ and $T_{min}$. The Yamamoto and Mann–Kendall tests indicated good consistency of the abruption test results over time; i.e., the abruption probability in S1 was higher than that in S2, and the abruptions occurred mainly in nighttime trends in S1 and daytime trends in S2. The S1 study area showed an abruption trend in $T_{min}$ from 2006 to 2017, and S2 also showed an abruption trend in $T_{max}$ from 2005 to 2011; both study areas experienced upward trends. This showed that the duration of warming abruption of the extreme $T_{min}$ values in S1 was longer than that of the extreme $T_{max}$ values in S2, and the duration of warming abruption in S1 was twice as long as that of S2 for extreme $T_{max}$ values. At the same time, this result indicated that the temperature difference of DNW in Xinjiang would show a decreasing trend in the future.

### 4.3. Effects of Asymmetric Warming on Vegetation NDVI

The NDVI is an indicator of vegetation growth change, and the responses of NDVI to $T_{max}$ and $T_{min}$ reflect the effects of DNW asymmetric warming on terrestrial ecology. However, the DNW was asymmetric, i.e., the effect of nighttime (0.65 °C (decade)$^{-1}$) warming on vegetation activity was more significant than that of daytime warming. A possible reason for this result is that plants fix carbon (C) by photosynthesis during the day, while respiration takes place all the time (day- and nighttime). The imbalanced DNW rate leads to an increase in the respiration of plants at nighttime and prompts an increase in the soil organic matter decomposition rate [25]. The growth of vegetation must rely on environmental temperatures for energy to regulate internal biochemical processes and further regulate growth in the phenological period. Nemani et al. [26] found that the limiting factor for vegetation growth in high-latitude and high-altitude areas was temperature. High-altitude vegetation was more sensitive to the asymmetry of DNW; this result was in line with the results obtained in the current study. Xinjiang is located in an arid and semiarid area, and water shortage is the main limiting factor for vegetation growth. The NDVI values were positively correlated with $T_{max}$ and $T_{min}$ in S1 (Junggar Basin desert steppe) and S2 (Taklimakan Desert basin); this correlation is largely attributable to the physiological regulation mechanism of vegetation, which enhances the water absorption capacity of plant roots experiencing drought stress. Peng et al. [22] suggested that DNW both enhanced and inhibited vegetation growth through different disturbance mechanisms. The western area of S1 has sufficient precipitation compared with that of S2; $T_{max}$ warming has a more significant positive effect on coniferous forests, broad-leaved forests, and meadow grasslands. When $T_{max}$ warms continuously and exceeds the optimal temperature required for vegetation growth, vegetation growth will be inhibited; previously, vegetation growth was mainly limited by precipitation. In addition, continuous warming of $T_{max}$ accelerates the evaporation of soil water in the daytime and aggravates the drought stress of the soil. In this situation, the vegetation cannot obtain the necessary moisture from the soil; therefore, the response of vegetation growth to increasing $T_{max}$ in S1 was mostly negatively correlated, and the response of grassland vegetation to increasing $T_{max}$ was significant. In contrast, S2 has scarce precipitation all year, a dry climate, and a high-altitude vegetation water supply that mostly comes from alpine meltwater. The NDVI was weakly correlated with $T_{max}$ in S2. To better explain the response mechanisms of vegetation activity and vegetation growth stress to extreme climate events and climate change, human activities and vegetation physiological characteristics should be comprehensively considered in future research.

## 5. Conclusions

This study analyzed monthly $T_{max}$, $T_{min}$, precipitation, and NDVI datasets (2000–2020) and determined the change trends of DNW, seasonal warming, and the diurnal temperature range in S1 and S2. The conclusions are as follows.

During the study period between 2000 and 2020, the DNW in S1 and S2 of Xinjiang showed warming trends, and the warming rates were higher than the world average. However, DNW, seasonal warming, and the diurnal temperature range in S1 and S2 were asymmetrical, and the nighttime warming rate (0.65 °C (decade)$^{-1}$) was faster than the daytime warming rate (0.4 °C (decade)$^{-1}$). The S1 area had higher nighttime warming trends than daytime warming trends in spring, summer, and winter, while the opposite trend was identified in autumn. The nighttime warming trend in S2 in spring and autumn was higher than the daytime warming trend, and the opposite trend was observed in winter. The interannual diurnal temperature range showed a trend of spring > autumn > summer > winter. The positive partial correlation between NDVI and $T_{min}$ was significantly higher than that between NDVI and $T_{max}$ in the area where the significance test passed. The abruption probability in S1 was higher than that in S2, and the abruptions mainly occurred in nighttime warming in S1 and daytime warming in S2. The S1 area showed an abruption trend in $T_{min}$ from 2006 to 2017, and S2 also showed an abruption trend in $T_{max}$ from 2005 to 2011; both exhibited upward trends. The responses of vegetation activities to DNW in Xinjiang had significant spatial differences, especially between the Junggar Basin (S1) and Gurbantünggüt Desert plain (S2).

**Author Contributions:** Formal analysis, data curation, writing—original draft, conceptualization, methodology, T.H.; conceptualization, methodology, writing, revision, supervision, G.F.; editing, Y.O.; editing, X.H. All authors have read and agreed to the published version of the manuscript.

**Funding:** This research was funded by the National Natural Science Foundation of China [grant number 51969027, U1803244] and the China Scholarship Council.

**Acknowledgments:** The authors are grateful to the anonymous reviewers and the corresponding editor for their helpful and constructive comments and suggestions for improving the manuscript.

**Conflicts of Interest:** The authors declare no conflict of interest.

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
