# Peer review of "The Spatiotemporal Patterns of Climate Asymmetric Warming and Vegetation Activities in an Arid and Semiarid Region"

_climate, doi:10.3390/cli8120145_

Round 1

Reviewer 1 Report

In my opinion the paper is well organized and presented. Genarally, it is a well-written and easy enough for the reader.

Some minor comments.

I would like to see more recent papers dealing with global warming rate in China (line 40).

Please correct syntax:

Lines 89-92

Line 98

Correct the word "analyzed" -->line 305

Thank you

Author Response

Response to Reviewer 1 Comments

Point 1: I would like to see more recent papers dealing with global warming rate in China (line 40).

Response 1: I have added references. [16-20]

Point 2: Please correct syntax: Lines 89-92, Line 98. Correct the word "analyzed" -->line 305

Response 2: I have revised as requested.

Reviewer 2 Report

The authors investigated the spatiotemporal patterns of asymmetric warming in Xinjiang to provide a reference for impacts of extreme meteorological events on the vegetation ecosystem in the studied region. Least squares linear regression, Yamamoto and Mann-Kendall nonparametric randomization tests, and second-order partial correlation analysis were applied to conduct trend analysis. The manuscript topic seems interesting but some sections, particularly the “Introduction” section, needs to be carefully improved. I recommend the paper for publication after major and minor revisions described below are addressed.

Major Comments:

  • The abstract should begin with a very brief introduction and indication of the importance of the studied topic rather than jumping into research results.
  • Lines 43 to 53- Please improve the “Introduction” section by providing in-depth discussions of the reviewed literature. I can see that only one short sentence is mentioned for each reviewed research. The readers need to also see their findings/applied methods to identify the importance and, specifically, difference of your research.
  • Line 85 – No need to have a separate section for “Data Sources”.
  • Line 102 – The Mann-Kendall and Yamamoto tests should be explained with details. For example, the Sign function that is used for Man-Kendall test should be discussed in the “Research Methods” section.
  • Line 150 - How authors can explain such opposite trend in S1 and S2? What factors/reasons can lead to such opposite trend in these two regions?
  • Key recommendations can be also reported in the last paragraph of the “Conclusions” section.

Minor Comments:

  • Line 17 – What is “P” referring to?
  • Line 43 – What does IPCC stand for? You need to mention that it stands for the Intergovernmental Panel on Climate Change
  • Line 48- Although the long form of NDVI has been mentioned in the abstract, the long form should be also used over the first appearance of the term in the text. Please revise the entire text accordingly.
  • Line 75- Replace “km2” by “km2
  • Line 77 – The reference (Feng et al., 1989) should be numbered.
  • Line 81 – Remove the first “°C”
  • Line 99 - What does “d” referring to? If it is referring to degree, you need to bring the term degree here.
  • Line 100- If MVC stands for Maximum Value Synthesis, then this abbreviation should be MVS.
  • Line 109 – “was” should be replaced by “is”
  • Line 166- The vertical axis of all plot should have the same scale. For example, they should all range between -30 to 34°C to be consistent for comparison. Please also revise Figures 4 and 6 accordingly.

Author Response

Response to Reviewer 1 Comments ( "Please see the attachment." ï¼‰

Major Comments:

Point 1: The abstract should begin with a very brief introduction and indication of the importance of the studied topic rather than jumping into research results.

Response 1: Asymmetric warming was bound to have a major impact on terrestrial ecosystems in arid regions during global warming. Further study was necessary to reveal the spatiotemporal patterns of asymmetric warming in Xinjiang.

Point 2: Lines 43 to 53- Please improve the “Introduction” section by providing in-depth discussions of the reviewed literature. I can see that only one short sentence is mentioned for each reviewed research. The readers need to also see their findings/applied methods to identify the importance and, specifically, difference of your research.

Response 2: Davy et al. [21] showed that the warming rate in summer is faster than those in spring and autumn in the high latitudes of the Northern Hemisphere. As an important part of terrestrial ecosystems, the growth and development of vegetation is bound to be affected by asymmetric warming. These generally relate differences in the temperature trends to regionalized cloud cover, precipitation or soil moisture. Peng et al. [22] analyzed the interannual covariations of the satellite-derived NDVI with Tmax and Tmin over the Northern Hemisphere, they study showed that daily maximum temperatures were positively correlated with the NDVI in humid areas and had a significant negative correlation with the NDVI in arid areas. Guang et al. [23] investigated vegetated growth dynamics (annual productivity, seasonality and the minimum amount of vegetated cover) in China and their relations with climatic factors during 1982-2011, they believed that vegetation productivity was positively correlated with the nighttime warming rate. Nemani et al. [24] analyzed the climatic data and satellite observations of vegetation activity during 1982-1999, and found that vegetation growth representing the response cycle of asymmetric warming in high-latitude and high-altitude areas was shorter than that in low-latitude and low-altitude areas. Cong et al. [25] showed that asymmetric warming could accelerate respiration in plants and increase the decomposition rate of organic matter. However, little is known about the warming trends in the relationships between NDVI and asymmetric warming and precipitation, and understanding this is crucial for predicting how climate change would affect vegetation activity in the future.

Point 3: Line 85 – No need to have a separate section for “Data Sources”.

Response 3: It has been combined with 2.1

Point 4: Line 102 – The Mann-Kendall and Yamamoto tests should be explained with details. For example, the Sign function that is used for Man-Kendall test should be discussed in the “Research Methods” section.

Response 4:The major mathematical methods are as follows:

For the time series, a rank s series is constructed:

                                      (2)

                                           (3)

Where, rank series  is the cumulative number of the  time value greater than j time. Define statistics under the assumption that the time series are independent:

                                     (4)  

                              (5)

 Where, ,  and  are the mean and variance of . When  are independent of each other and have the same continuous distribution, they are calculated as follows:

                          (6)

Set  and  as standard normal distribution, given a significance value α=0.05, critical value U0.05=±1.96, and Two statistical sequence curves of   and  and the two critical lines (±1.96) were drawn in one graph.

Yamamoto tests

The specific method is to define a signal-to-noise ratio:

                                                              (7)

Where,  is the population sample variance, and  is the length of the population sample sequence; and  are average values of 2 sub-sample sets;   and  are Variance of sub-sample. Molecular was defined when  

Point 5: Line 150 - How authors can explain such opposite trend in S1 and S2? What factors/reasons can lead to such opposite trend in these two regions?

Response 5: while the temperatures in S2 showed a cooling trend. This phenomenon is mainly due to the latitude differences between S1 and S2, which further supports the significant warming trend in higher latitudes in the northern hemisphere.

Minor Comments:

Point 6: Line 17 – What is “P” referring to?

Response 6: Indicated significant, .

Point 7: Line 43 – What does IPCC stand for? You need to mention that it stands for the Intergovernmental Panel on Climate Change

Response 7: I have added.

Point 8: Line 48- Although the long form of NDVI has been mentioned in the abstract, the long form should be also used over the first appearance of the term in the text. Please revise the entire text accordingly.

Response 8: I have added in the introduction.

Point 9: Line 75- Replace “km2” by “km2

Response 9: I have revised as requested.

Point 10: Line 77 – The reference (Feng et al., 1989) should be numbered.

Response 10: I have revised as requested.

Point 11: Line 81 – Remove the first “°C”

Response 11: I have revised as requested.

Point 12: Line 99 - What does “d” referring to? If it is referring to degree, you need to bring the term degree here.

Response 12: "D" means the number of days, which has been modified to 30 days.

Point 13: Line 100- If MVC stands for Maximum Value Synthesis, then this abbreviation should be MVS.

Response 13: I have revised as requested.

Point 14: Line 109 – “was” should be replaced by “is”

Response 14: I have revised as requested.

Point 15: Line 166- The vertical axis of all plot should have the same scale. For example, they should all range between -30 to 34°C to be consistent for comparison. Please also revise Figures 4 and 6 accordingly.

Response 15: I have tried it again, but the effect is not very good (as shown below), it is better to follow the original layout. Do you think I can only modify the colors of S1 and S2 without modifying the vertical axis?

Reviewer 3 Report

This paper is discussing the spatiotemporal pattern of Tmax and Tmin in an arid and semiarid region, and the impact of the temperature pattern on vegetation. The findings of this study are giving insight on the inter-relation between local climate change and ecosystem, particularly the vegetation.

General Part

Figure 2 should has different colour for S1 and S2. The legend should clear enough.

Figure 3 is a bit confusing. It is not easy to make a different between S1 and S2. Please give an appropriate legend for the graph.

Figure 8 is better to have the same scale of the legend

The conclusion part is much better and interesting if it is written in an explanatory way instead of a number of lists. I suggest rewriting the conclusion part.

Specific Part

40-42   Please give supporting evidences for the sentence about the significant impacts of asymmetric warming on terrestrial vegetation …….

139 – 141        This paragraph discusses the general trend of Tmin and Tmax in S1 and S2, but the closing sentence says something about the day and nighttime temperature warming without providing the quantitative results. Hence I suggest providing the rates to support the statement.

145 – 152        You mention that Tmax in S1 showed a cooling trend in summer and significant trends in other seasons. According to Figure 3, the Tmax in S1 had an increasing trend in Winter and a cooling trend in other seasons.  The contradiction is also found for Tmax in S2. Why are they different?

153 – 164        This paragraph might be confusing since the explanation is given in an unstructured way. For example, the warming rates comparison of Tmax and Tmin is carried out between two seasons in two different areas, S1 and S2. I suggest starting the explanation by comparing the warming rates for each season and then comparing two different seasons.

153 – 154        what do you mean by significant?.

186 – 187        what is the proportion you are talking about?

223 – 237        can we have two separate paragraphs that discuss the partial correlation in S1 and S2 of Tmax and Tmin?

254 -259          Can you provide some examples from other areas with similar climate condition to support this statement?

259 – 260        do you have any references to support this statement?

Author Response

Point 1: Figure 2 should has different colour for S1 and S2. The legend should clear enough.

Response 1: I have modified Figure 2.

Point 2: Figure 3 is a bit confusing. It is not easy to make a different between S1 and S2. Please give an appropriate legend for the graph.

Response 2: I have revised Figure 3.

Point 3: Figure 8 is better to have a same scale of the legend.

Response 3: Figure 8 (a) and (b) have specific partial correlation coefficient thresholds.

Point 4: The conclusion part is much better and interesting if it is written in an explanatory way instead of a number of lists. I suggest to rewrite the conclusion part.

Response 4: I have rewritten it.

Point 5: 40-42 Please give supporting evidences for the sentence about the significant impacts of asymmetric warming on terrestrial vegetation.

Response 5: I have added references in the Discussion section (Response 12).

Point 6: 139 141 This paragraph discusses the general trend of Tmin and Tmax in S1 and S2, but the closing sentence says something about the day and nighttime temperature warming without providing the quantitative results. Hence I suggest to provide the rates to support the statement.

Response 6: I have revised as requested.

Point 7: 145 152 You mention that Tmax in S1 showed a cooling trend in summer and significant trends in other seasons. According to the Figure 3, the Tmax in S1 had an increasing trend in Winter and a cooling trend in other seasons. The contradiction is also found for Tmax in S2. Why are they different?

Response 7: I have revised as requested. in these three seasons, the Tmax and Tmin in S1 both showed a warming trend, while the temperatures in S2 showed a cooling trend. This phenomenon is mainly due to the latitude differences between S1 and S2, which further supports the significant warming trend in higher latitudes in the northern hemisphere.

Point 8: 153 164 This paragraph might be confusing since the explanation is given in an unstructured way. For example, the warming rates comparison of Tmax and Tmin is carried out between two seasons in two different areas, S1 and S2. I suggest starting the explanation by comparing the warming rates for each season and then comparing two different seasons.

Response 8: I have rewritten it.

Point 9: 153 154 what do you mean by significant?

Response 9: I have rewritten it.

Point 10: 186 187 what is the proportion you are talking about?

Response 10: I have rewritten it. The Tmax and Tmin in S2 showed cooling trends in the southwest and central regions, and the proportions of cooling trends were 5.3% and 2.9%, respectively.

Point 11: 223 237 can we have two separate paragraphs that discuss the partial correlation in S1 and S2 of Tmax and Tmin?

Response 11: I have revised as requested.

Point 12: 259 260 do you have any references to support this statement?

Response 12: I have added references. [44-47]

Reviewer 4 Report

The paper presents a study of spatial and temporal changes of the DNW in Xinjiang province, China, for a period of 20 years (2000- 2020). The authors analyzed temperature data from 34 meteorological stations and NDVI from remote sensing, finding that DNW was asymmetric, and examined the impact of DNW on the vegetation in terms of NDVI. Although the study has a regional character, it is of interest to a wide audience, considering the arid and semiarid nature of the region.

The methods of analysis used in the study (Least squares linear regression, Yamamoto and Mann-Kendall nonparametric randomization tests, and second-order partial correlation) were applied correctly, but some of them require more details.

The authors claim “The goal of this study is to provide a reference for the impacts of extreme meteorological events on the vegetation ecosystems in Xinjiang“, Furthermore, the title of the article notes that the present study examines: “The spatiotemporal patterns of climate asymmetric warming and vegetation activities in an arid and semiarid region“. However, the authors dedicate very little attention to the impact of asymmetric warming on vegetation, which is presented only in section „3.4. Partial correlation analysis between vegetation activity and asymmetric DNW in Xinjiang“. To justify the purpose of the study, as well as the title of the article, the authors need to perform an in-depth analysis of the relationship between DNW and vegetation development. Is there a difference in the vegetation development in arid and semiarid area with respect to DNW? How does DNW impact different ecosystems (i.e. mountain-oasis-desert) and different vegetation (i.e. forest, grassland, shrubland, wetlands, etc)?

The authors should redefine the purpose of the study. They state that “The goal of this study is to provide a reference for the impacts of extreme meteorological events on the vegetation ecosystems in Xinjiang“. However, no extreme meteorological events have been presented in the study. The results show that the main goal is to study DNW in the region and its impact on vegetation in terms of NDVI. Maximum and minimum temperatures are not extreme meteorological events.

Precipitation is used as a control variable in the partial correlation analysis of NDVI and DNW, as a characteristic of the studied regions, as well as in the discussion of the impact of warming on vegetation.. Please specify what is the impact of precipitation on the partial correlation between NDVI and DNW.

The authors use the term “mutation” in DNW analysis: “mutation of temperature”, “mutational signatures”, “mutation characteristics” etc., which are not defined. The term “mutation” is specific to biology (genetics). When applying the M-K and Yamamoto tests, the authors should use the established terminology: abruption, points of abrupt, abrupt changes, discontinuous points, etc.

The charts are cluttered with a lot of information. To improve the presentation, consider presenting certain data, such as regression equations and explanatory text, separately in tables or in the title of the chart. Chart titles need revision because they contain inaccuracies or are unclear.

Replace "diurnal range" in the text with "diurnal temperature range".

As a reader, I struggled with this paper as statements are often grammatically incorrect and certain sentences are long and hard to follow. The terminology used by the authors to describe the methods and results of analysis is also rather uncommon.  Although the results of the study are correct, their presentation is often confusing and unclear. The conclusions are presented chaotically and need to be redefined.

The authors need to put more effort in the presentation and explanation of the results of their research. The manuscript requires re-writing. I recommend consultation with a professional English translator to help with the structure of the entire paper.

Considering the remarks noted above and the specific comments, I recommend that this manuscript undertakes deep revision before being considered for publication.

Round 2

Reviewer 2 Report

Fortunately, the authors have addressed all comments properly. My concern on in-depth discussions of the reviewed literature have been addressed. Regarding my last comment (point 15), I leave it to authors to decide on their preference for vertical axis scaling.

Overall, I recommend the manuscript for publication.

Reviewer 3 Report

This manuscript has been improved. The authors have considered all feedback that had been given in the first review. The figures are now clearer which can explain the results graphically. Some sections are more structured than before, particularly in the conclusion section.

Congratulation

Reviewer 4 Report

The authors have correctly revised the content of the manuscript. I recommend the manuscript for publishing in Climate Journal.